# Comprehensive RNA-Seq Analysis Pipeline for Non-Model Organisms and Its Application in *Schmidtea mediterranea*

**DOI:** 10.3390/genes14050989

**Published:** 2023-04-27

**Authors:** Yanzhi Wang, Sijun Li, Baoting Nong, Weiping Zhou, Shuhua Xu, Zhou Songyang, Yuanyan Xiong

**Affiliations:** 1Key Laboratory of Gene Engineering of the Ministry of Education, Institute of Healthy Aging Research, School of Life Sciences, Sun Yat-sen University, Guangzhou 510006, China; 2Guangdong Provincial Key Laboratory of Malignant Tumor Epigenetics and Gene Regulation, Breast Tumor Center, Sun Yat-sen Memorial Hospital, Sun Yat-sen University, Guangzhou 510006, China; 3Maternal and Child Health Research Institute, Translational Medicine Center, Guangdong Women and Children Hospital, Guangzhou 511400, China; 4School of Life Sciences, Fudan University, Shanghai 200433, China

**Keywords:** non-model organisms, RNA-seq analysis, pipeline

## Abstract

RNA sequencing (RNA-seq) is a high-throughput technology that provides in-depth information on transcriptome. The advancement and dropping costs of RNA sequencing, accompanied by more available reference genomes for different species, make transcriptome analysis in non-model organisms possible. Current obstacles in analyzing RNA-seq data include a lack of functional annotation, which may complicate the process of linking genes to corresponding functions. Here, we provide a one-stop RNA-seq analysis pipeline, PipeOne-NM, for transcriptome functional annotation, non-coding RNA identification, and transcripts alternative splicing analysis of non-model organisms, intended for use with Illumina platform-based RNA-seq data. We performed PipeOne-NM on 237 *Schmidtea mediterranea* RNA-seq runs and assembled a transcriptome with 84,827 sequences from 49,320 genes, identifying 64,582 mRNA from 35,485 genes, 20,217 lncRNA from 17,084 genes, and 3481 circRNAs from 1103 genes. In addition, we performed a co-expression analysis of lncRNA and mRNA and identified that 1319 lncRNA co-express with at least one mRNA. Further analysis of samples from *S. mediterranea* sexual and asexual strains revealed the role of sexual reproduction in gene expression profiles. Samples from different parts of asexual *S. mediterranea* revealed that differential expression profiles of different body parts correlated with the function of conduction of nerve impulses. In conclusion, PipeOne-NM has the potential to provide comprehensive transcriptome information for non-model organisms on a single platform.

## 1. Introduction

RNA sequencing (RNA-seq) can profile the whole transcriptome and potentially lead to the identification of novel transcripts. According to their protein-coding ability, transcripts can be divided into protein-coding transcripts and nonprotein-coding transcripts. Both play important roles in cellular processes. Protein-coding transcripts, mRNA, serve as translation templates for all peptides and proteins in the cell. Some protein-coding transcripts go through one or more forms of alternative splicing before maturation. Alternative splicing is an essential component of gene expression regulatory systems, and different isoforms from the same gene can be involved in different cellular processes. Nonprotein-coding transcripts include long noncoding RNAs (lncRNAs), circular RNAs (circRNAs), and microRNAs (miRNAs). lncRNAs can regulate cellular processes by (1) interacting with other components in the cell, (2) using their gene regulatory element to direct the activity of transcription regulatory element, or (3) influencing gene activity through their transcription process [1]. circRNAs are covalently closed RNA generated by back-splicing, acting as microRNA sponges, RNA-binding protein sequestering agents, and nuclear transcriptional regulators in cells [2]. Based on the origin of circRNA, they can be categorized as circRNAs, which consist of exons (ecircRNAs), or circRNAs, which consist of introns (ciRNAs).

The advances and decreasing costs of next-generation sequencing contribute to genome assembly for more and more non-model organisms, which enables transcriptome analysis to discover novel genes or transcripts. Unlike model organisms, RNA-seq analysis of non-model organisms is faced with unique challenges, including a lack of functional annotations. Without functional gene annotations, exploring the function of differentially expressed genes and enriching these genes in certain pathways can be complicated. Some packages have been developed to carry out these functions already, focusing on effectively assembling and annotating genome of non-model organisms [3,4]. With functional annotation of genome and transcriptome, studies of non-model organisms can help shed light on gene expression regulatory networks under different biological contexts [5], thus utilizing some of the key components in order to bring economic or ecological benefits. Besides, numerous studies have expanded their scope of research to non-protein coding transcripts, exploring the function of non-coding RNAs in the growth and development of non-model organisms [6]. However, there is still a lack of RNA-seq analysis tools that integrate transcriptome annotation and exploration for non-model organisms. By studying transcriptomes with comprehensively functional annotation, we can achieve a deeper understanding of the regulatory networks of transcripts in non-model organisms under different biological contexts.

*Schimidtea mediterranea* is an ideal model species for regenerative biology, developmental biology, and stem cell biology research. In recent years, a large amount of RNA-seq data have been generated, but each study usually focused on only one or two aspects. We focus on data collection and integration analysis and bring together multiple RNA-seq data samples published by different researchers to obtain comprehensive transcriptional information of *S. mediterranea* under different biological conditions.

In this article, we introduce PipeOne-NM, a comprehensive pipeline for RNA-seq analysis of non-model organisms. The pipeline is applicable to Illumina platform-based RNA-seq datasets with an available reference genome. The pipeline can be applied in pre-processing raw RNA-seq datasets, reconstructing transcriptomes, and assembling genome annotations. Moreover, it can quantify transcripts, perform alternative splicing analysis, and predict transcript function based on base content and homology protein searches in databases. The downstream analysis module also identifies lncRNA and circRNA in transcriptomes and enriches differentially expressed genes in pathways. Together, the pipeline helps to comprehensively annotate and analyze transcriptomes of non-model organisms. We apply PipeOne-NM to 237 RNA-seq runs for *S. mediterranea*. PipeOne-NM is freely available at https://github.com/Lisijun-m/pipeone-nm (accessed on 20 February 2023).

## 2. Methods and Materials

### 2.1. RNA-Seq Data Collection

We collected 237 Illumina platform-based, bulk RNA-seq runs of *Schmidtea mediterranea* from the SRA database of the NCBI. Of all the 237 RNA-seq runs, 63 runs were from a sexual strain and 174 from an asexual strain, covering 120 biological treatments. All data were downloaded via prefetch from the SRA toolkit (version 2.11.2) (http://ncbi.github.io/sra-tools, accessed on 23 February 2023). We also collected 6 LS454 platform-based single-end RNA-seq runs of an asexual strain for the de novo construction of a reference transcriptome. Additionally, 160,721 and 78,586 sequencing reads of RNA from the Nucleotide and Nucleotide EST databases of the NCBI were collected for de novo transcriptome construction (See in Appendix A).

### 2.2. Sequence Alignment and Transcriptome Assembly

A standardized RNA-seq analysis pipeline, PipeOne-NM was performed for 237 downloaded RNA-seq runs. Downloaded RNA-seq runs were format-converted via fasterq-dump from the SRA toolkit and then were quality-controlled using fastp (version 0.23.0) [7]. Specifically, given that there are sexual and asexual strains of *S. mediterranea*, quality-controlled RNA-seq runs were firstly aligned to the reference genome of *S. mediterranea* (sexual strain S2) [8] using HISAT2 (version 2.2.1) [9]; then, those unmapped reads were aligned to the *S. mediterranea* asexual strain CIW4 reference genome.

To increase the mapping rate of 237 RNA-seq runs, 6 LS454 platform-based single-end RNA-seq runs of the asexual strain and read sequences from the Nucleotide and Nucleotide EST databases of the NCBI were filtered by fastp and were used to de novo construct a reference transcriptome via Trinity (version 2.6.5) [10]. After two sets of transcriptomes had been integrated, TGICL (version 2.1) [11], CAP3 (version 02/10/15) [12], and CD-HIT-EST (version 4.8.1) [13] were used to assemble a reference transcriptome. Unmapped reads of the two above reference genomes were aligned against the reference transcriptome using HISAT2. For non-model organisms analyzed by PipeOne-NM, RNA-seq runs were only aligned to the reference genome once, with the generated alignment result being stored in a SAM format file.

The resulting SAM files were converted into sorted BAM files using SAMtools (version 1.13) [14]. The transcriptome of each RNA-seq run was reconstructed using StringTie (version 2.1.6) [15] and merged to create an annotation GTF file for *S. mediterranea* using TACO (version 0.7.3) [16].

N50 is a statistical indicator of the quality of assembly, and ExN50 (Expression-dependent N50) is calculated as the N50 length against an x fraction of the total expressed data (Ex). The ExN50s with x ranging from 0 to 100 were calculated to show the length distribution of expressed transcripts.

### 2.3. Transcript Quantification and Transcriptome Annotation

In PipeOne-NM, Salmon [17] was used to estimate the expression levels in Transcripts Per Million (TPM) in each sample. The quantitative results of all samples were summarized as an expression matrix, and the expression levels were normalized between samples using Trimmed Mean of M values (TMM) [18]. The transcripts were filtered, with the threshold of raw expression level ≥ 10, TPM ≥ 0.1, and TMM ≥ 0.1 in at least 1/20 of the runs (12 runs in 237 total runs). To measure the degree of tissue specificity of gene expression, we calculated the τ value [19] for each gene.

ORFs (open reading frames) of each transcript were extracted using TransDecoder (version 5.5.0) and the open reading frames were aligned in order against protein sequences of UniProt Swiss-Prot [20] and Pfam-A [21] by BLASTP (version 2.12.0) and hmmscan (version 3.3) [22]. Transcripts were aligned with protein sequences of UniProt Swiss-Prot by BLASTX (version 2.12.0). Signal peptide, trans-membrane domain, and rRNA transcripts were respectively predicted by SignalP (version 1.12) [23], tmhmm (version 2.0) [24], and RNAmmer (version 1.2) [25].

### 2.4. LncRNA Identification and LncRNA-mRNA Co-Expression Analysis

In the data analysis with PipeOne-NM, all transcripts were divided into three categories: rRNA, lncRNA, and mRNA. Transcripts with RNAMMER annotation were classified as rRNA; transcripts without BLASTX annotation and ORFs of length over 100 amino acids were classified as lncRNA; the rest were classified as mRNA, from which those transcripts with at least one of the two annotations, GO-BLAST and GO-Pfam, were selected for co-expression annotation of lncRNA.

Specifically, to validate the reliability of identified mRNAs, we downloaded the Unigene sequences of *S. mediterranea* from the SmedGD (https://planosphere.stowers.org/smedgd, accessed on 23 February 2023), including the nucleic acid sequence NT and the protein sequence AA. The nucleic acid sequence and protein sequence were used to construct the BLAST database and the mRNA sequences were aligned with sequences in the BLAST database.

For lncRNA-mRNA co-expression analysis, the Spearman correlation coefficient of the expression level of the lncRNA-mRNA pair with functional annotation was calculated. If the absolute value of the obtained correlation coefficient was greater than 0.8 and the corrected *p* value was less than 0.1, the expression levels of the lncRNA-mRNA pair were considered to be significantly correlated. Co-expressed mRNAs and the corresponding genes were used to perform GO analysis by R package goseq, and the over-represented FDR values were extracted from the results, while GO entries with less than 0.05 were considered as annotations associated with that lncRNA.

### 2.5. CircRNA Detection

In PipeOne-NM, BWA (version 0.7.17) [26] was used to align each fastq file to a reference genome, and CIRI2 (version 2.0.6) [27,28], CIRI-AS (version 2.0.6), and CIRIquant (version 1.0) [29] were used to detect circRNA, analyze the alternative splicing events and quantify the circRNA host gene count in each run. The fragments per kilobase of transcript per million mapped reads (FPKM) and TPM of circRNA host genes were obtained. GO enrichment analysis was performed by R package goseq.

### 2.6. Differential Expression and Gene Ontology Analysis

In the analysis of *S. mediterranea*, differentially expressed genes were identified using Trinity script run_DE_analysis.pl. For RNA-seq runs with biological replications, DESeq2 [30] was used to perform differential analysis; for runs without duplicates, edgeR [31,32] was used with a dispersion parameter of 0.1. Genes with a threshold of more than 4-fold change in expression levels between the two groups of samples and *p* < 0.001 were considered as differentially expressed genes. We postulate that the functions of certain gene products are similar with those of homologous gene products in protein databases, so the Gene Ontology annotation construction of non-model organisms was based on the BLAST results against UniProt Swiss-Prot and Pfam-A. In preparation for the gene enrichment analysis, scripts built in PipeOne-NM extracted and integrated transcriptome annotations, which were generated from Uniprot and Pfam-A BLAST results, as a file containing mapping the relationships between Gene Ontology terms and Gene Ontology ids and between Gene Ontology ids and gene ids. The mapping relationships could then be imported in the downstream Gene Ontology analysis. GO enrichment analysis was performed by R package goseq for the differential expressed genes, and an enriched list of GO entries sorted by over-represented FDR values was generated. GO enrichment data were visualized by the R package GOplot [33].

## 3. Results

### 3.1. Overview of PipeOne-NM Workflow

PipeOne-NM is a comprehensive RNA-seq analysis workflow for RNA-seq data of non-model organisms which offers a one-stop solution for genome annotation and transcriptome exploration. Based on Nextflow and Docker, installation and management of the workflow are easy. For users that may not have root permission (i.e., cannot pull the docker image), using Conda to install can circumvent the tedious installation, configuration, and management. PipeOne-NM aggregates and coordinates 23 tools to fully extract and analyze transcriptome information. Featuring an analysis pipeline for non-model organisms, PipeOne-NM tackles the problem of lacking species functional annotations and manages to establish a comprehensive transcriptome analysis system that works efficiently. PipeOne-NM is suitable for non-model organisms without fully annotated genomes or transcriptomes. For fully annotated species, we recommend using PipeOne [34] for relevant analyses.

PipeOne-NM is applicable to RNA-seq data downloaded from GEO database and raw sequencing data provided by users (Figure 1). In the upstream analysis, PipeOne-NM integrates seven different tools to pre-process RNA-seq data and reconstruct the transcriptome. Starting from the preprocessing of raw data downloaded from GEO database, PipeOne-NM utilizes tools that suit non-model organisms to provide trustworthy results. Current studies often collect and analyze multiple samples of non-model organisms in one project; the huge number of samples poses a greater challenge to the accuracy of transcriptome assembly combinations. PipeOne-NM tailors the analysis pipeline for projects with multiple samples by adopting the transcriptome-merging tool TACO, which will correctly delineate transcript start and end sites and assemble transcripts from a network of splicing patterns [16]. Reliable transcriptome combination will increase the credibility of the results in downstream analyses. Then, PipeOne-NM utilizes fifteen different tools to achieve novel transcript identification, alternative splicing analysis, transcript expression quantification, and gene enrichment analysis and visualization. Given that the potential huge samples analyzed in one project may increase analysis time greatly, PipeOne-NM chooses the highly efficient transcript quantification tool, Salmon [17], which quantifies transcript expression with raw reads, and the circRNA identification tool, CIRI2 [27], which is highly efficient when dealing with multiple samples. As for the alternative splicing analysis of non-model organisms, PipeOne-NM offers two choices, rMATS [35] or ASGAL [36], to allow a comprehensive alternative splicing analysis of transcripts.

The high-performance analysis tools process the data in a relatively shorter time and provide credible analysis results for a large number of samples in non-model organism projects. The results of transcriptome analysis can be further input in gene differential expression analysis and gene ontology analysis in R; the relevant tools are listed in Table 1.

Compared with the previous RNA-seq pipeline for non-model organisms, FA-nf [4], PipeOne-NM integrates predictions for both novel lncRNAs and circRNAs, retrotranscriptomes, and alternative splicing (Table 2).

### 3.2. Sequence Alignment and Transcriptome Assembly

Six LS454 platform-based *S. mediterranea* RNA-seq runs and reads from Nucleotide and Nucleotide EST databases were first separately de novo assembled into 38,780 and 31,085 transcripts. The two kinds of reference transcripts were integrated into one reference transcriptome containing 62,562 transcripts. The GC content of the reference transcriptome was 33.73%, which was higher than the GC content of the S2 strain reference genome (29.70%) (Table 3).

The mapping rate of the Illumina platform-based RNA-Seq data, which were first aligned against the sexual S2 strain reference genome, was nearly 90% for most sexual runs and 75–90% for asexual runs. Unmapped reads of the S2 sexual strain reference genome were aligned against the asexual CIW4 strain reference genome and assembled reference transcripts, and the mapping rates of most runs were above 50%.

The complete transcriptome for expression estimation was integrated from transcripts of each Illumina platform-based RNA-seq run, containing a total of 107,317 transcripts. The complete transcriptome for annotation retained sequences with expression raw value ≥ 10, TPM ≥ 0.1, and TMM ≥ 0.1 in at least 12 samples (1/20 of 237 runs); for multiple transcripts corresponding to the same gene, those with expression levels less than 1% of the highest were filtered, resulting in a total of 84,827 sequences from 49,320 genes (see in Methods and Materials). The GC content of the transcriptome for annotation was 34.37%, i.e., slightly higher than that in the S2 strain reference genome (29.70%) (Table 4).

The ExN50s with x ranging from 0 to 100 were calculated to show the distribution of the lengths of transcripts. From 14% to 82%, the N50 lengths of the top expressed transcripts were below 400; even in the top 90% of expressed transcripts, the N50 length was only 529. After over 90% of the top expressed transcripts were included, the curve rose sharply, and finally, the length of N50 (E100N50) reached 1501. The change of length of N50 in *S. mediterranea* indicated that longer transcripts tended to be expressed at a lower level while shorter transcripts tended to be expressed at a higher level (Figure 2A).

### 3.3. Transcript Quantification and Transcriptome Annotation

All assembled transcripts were classified into three categories: rRNA, lncRNA, and mRNA. Next, 28 transcripts with RNAMMER annotation were annotated as rRNAs. Meanwhile 20,217 transcripts from 17,084 genes were annotated as lncRNA, based on the threshold of no BLASTX annotation and open reading frames of over 100 amino acids. The remaining 64,582 transcripts with BLASTX annotation from 35,485 genes were annotated as mRNAs. Compared with the number of genes, a larger number of transcripts indicated that alternative splicing occurred both in lncRNA and mRNA. The total gene number of *S. mediterranea* is 49,320, so it can be postulated that 3269 genes code both coding and non-coding transcripts, while the remaining genes produce only one of the two (Figure 2B).

The maximum length of mRNA is greater than lncRNA, indicating that the length of mRNA is greater in *S. mediterranea*. However, the GC content was closer between the lncRNA and mRNA; both were slightly higher than that of the S2 strain reference genome (29.70%) (Table 5). The length of lncRNA is below 1000 bp, with 250–300 bp being the most abundant; the length of mRNA is between 300–2000 bp (Figure 2C). As for the validation of mRNAs, more than 99% of the 64,582 mRNA sequences identified in this study could be aligned to the Unigene nucleic acid and protein sequences in SmedGD (Table 6).

### 3.4. LncRNA-mRNA Co-Expression Analysis and CircRNA Detection

Spearman correlation coefficients were calculated between 45,578 identified mRNAs with at least one annotation of GO-BLAST or GO-Pfam and lncRNA, resulting in 10,185 pairs of significantly co-expressed lncRNA-mRNA involving 1319 lncRNA. Among all the co-expressed genes for gene enrichment analysis, the co-expressed genes of 1300 lncRNAs obtained GO enrichment results. GO entries with FDR less than 0.05 were extracted from the enrichment results as annotations for the corresponding co-expressed lncRNAs, and a total of 340 lncRNAs were annotated.

A total of 3481 circRNAs from 1103 circRNA host genes, including 1308 ecircRNA from 647 genes and 2173 ciRNAs from 506 genes, were identified from the transcriptome. The number of circRNAs is higher than the number of genes, indicating that the alternative splicing which occurs during circRNA formation is more frequent in ciRNAs (Figure 2D). The GC content of both types of circRNAs was lower than the GC content of the reference transcriptome (34.37%) and reference genome (29.70%) (Table 7). Length distributions were similar between ecircRNAs and ciRNAs, clustering around 300 bp (Figure 2E).

GO enrichment analysis was performed separately for ecircRNA host genes. The FDR value was less than 0.001 for ecircRNA host genes. Additionally, among the 777 GO annotations with FDR ≤ 0.05, 107 involved membrane bound organelles, i.e., cytoplasmic components in the cellular component (MF) (Figure 3A); 81 involved protein binding in molecular functions (MF) (Figure 3B); 589 involved developmental processes, biological processes, and cellular processes and their regulation, positive regulation, and bioregulation in biological processes (BP). The annotation results were consistent with the molecular function of circRNAs as miRNA sponges.

### 3.5. Differential Expression Analysis of Schmidtea mediterranea

Differential expression analysis between mature asexual and mature sexual worms identified 6496 up-regulated genes in the asexual strain and 4994 up-regulated genes in the sexual strain. GO enrichment analysis was performed separately for these two groups of up-regulated genes. Ten GO annotations of up-regulated genes in the sexual strain, with FDR less than 0.001, involved microtubules, polymeric cytoskeletal fibers, and supramolecular polymers/fibers/complexes in cellular components (CC), cytoskeletal structural components and L-proline transmembrane transport carrier activities in molecular functions (MF), microtubule-based processes, proline transport and transmembrane transport in biological processes (BP). Among the GO annotations, microtubules and the cytoskeleton are involved in meiosis and sexual reproduction, indicating the differences between gene expression profiles of mature sexual and asexual worms (Figure 4A).

A differential expression analysis between mature sexual and juvenile sexual worms identified 4751 up-regulated genes in mature sexual worms and 277 up-regulated genes in juveniles, indicating that more genes were up-regulated in adult sexual strain *S. mediterranea*. GO enrichment analysis was performed for these two groups of genes separately, and the smallest FDR value obtained for the juvenile sexual worms was 0.1237. FDR was less than 0.001 for 28 GO-annotated up-regulated genes in mature sexual worms and was less than 1 × 10^−8^ in 9 GO annotations. These nine GO annotations involved microtubules, polymeric cytoskeletal fibers, and supramolecular polymers/fibers/complexes in cellular components (CC), cytoskeletal structural components and guanylase activity in molecular function (MF), and microtubule-based processes and proline transport in biological processes (BP). The above GO entries are similar with those GO annotations of genes with upregulated expression levels in mature worms of the sexual strain, suggesting that the up-regulated genes were involved in meiosis and sexual reproduction (Figure 4B).

Among the RNA-seq runs, PRJNA314467 divided six anterior–posterior “zones” for asexual strain in *S. mediterranea* [46]. Differential expression analysis was performed between these six groups of samples and asexual mature worm samples respectively, and up-regulated genes were selected for GO enrichment analysis. Notably, 79 genes in eye samples yielded a total of 53 GO annotation entries with FDR values less than 0.05, including 19 entries with FDR values less than 0.005. These 19 GO annotations did not involve cellular components (CC) but rather sodium channel activity, molecular and signaling activities, substrate specific channel activity, passive transmembrane carrier activity, transmembrane receptor activity, signaling receptor activity, and sodium transmembrane carrier activity in molecular functions (MF), as well as negative regulation of action potential, determination of liver left-right asymmetry, external/abiotic and other stimulus detection, and transport of monovalent inorganic positive ions such as sodium ions in the biological processes (BP). Because *S. mediterranea* do not have a liver, the annotation hepatic asymmetric decision indicates that the gene are homologous to genes performing this function in higher animals; however, the other MF and BP annotations are closely related to nerve impulse conduction, which is due to the forward concentration of nerve cells in vortex worms, forming the functional brain (Figure 4C).

## 4. Conclusions

In this study, a one-stop and comprehensive transcriptome analysis pipeline for non-model organisms was performed in data analysis of *S. mediterranea* RNA-seq data. Given that there are different strains of *S. mediterranea* (sexual and asexual strains), in order to improve the mapping rate of RNA-Seq data, the LS454 platform-based sequencing data and nucleic acid sequences were fully utilized to assemble a reference transcriptome. Then, the 237 Illumina platform-based RNA-Seq data samples from 120 biological conditions were successively aligned against two reference genomes and built reference transcriptomes, assembling into a transcriptome with an N50 length of 1501 and 84,827 sequences from 49,320 genes. A BLAST of the transcriptome sequences identified 64,582 mRNA and 20,217 lncRNA sequences in the transcriptome, respectively corresponding to 35,485 and 17,084 genes. Co-expression analysis of lncRNA and mRNA identified that 1319 lncRNA co-express with mRNA with at least one mRNA. Among unmapped reads against the reference genome and reference transcriptome, 3481 circRNA transcripts, of which 1308 consisted of exons, were identified, indicating the active role of circRNA in biological processes of *S. mediterranea*.

Differential expression and GO enrichment analysis were performed, reflecting the roles of genes in various biological process, including sexual reproduction, clustering of nerve cells near functional brain. The RNA-seq analysis results indicate that this transcriptome analysis pipeline is reliable and can be used to analyze RNA-seq data of more non-model organisms.

## 5. Discussion

The study demonstrates the feasibility of the RNA-seq pipeline, PipeOne-NM, for transcriptome analysis of non-model organisms. PipeOne-NM is a comprehensive RNA-seq analysis workflow for RNA-seq data of non-model organisms which offers a user-friendly solution to genome annotation and transcriptome exploration.

One limitation of PipeOne-NM is that it currently focuses on the transcriptome analysis of non-model organisms with reference genomes. Even though nowadays more and more reference genomes for multiple non-model organisms are becoming available, there is still a lack of an efficient computational method to perform sequencing analysis for a non-model organism from scratch, including de novo genome assembly, genome assembly evaluation, and later, transcriptome analysis. In the future, more functions will be implemented in PipeOne-NM to allow a multi-omics analysis of non-model organisms when this information is available. Although PipeOne-NM can provide a one-stop solution for the analysis of non-model organisms, the current PipeOne-NM still requires users to have some basic computational knowledge to interpret the analysis results. Future directions for improving PipeOne-NM could include providing more comprehensive and interactive reports of analysis results, so that more researchers can understand the analysis of non-model organism transcriptomes.

In short, we presented an analysis pipeline integrating various kinds of RNA-seq analysis tools to carry out transcriptome analysis for non-model organisms with a reference genome. Our pipeline provides new possibilities for transcriptome research of non-model organisms and may bring more attention to the value of non-model organisms in environmental and evolutionary studies.

## Figures and Tables

**Figure 1 genes-14-00989-f001:**
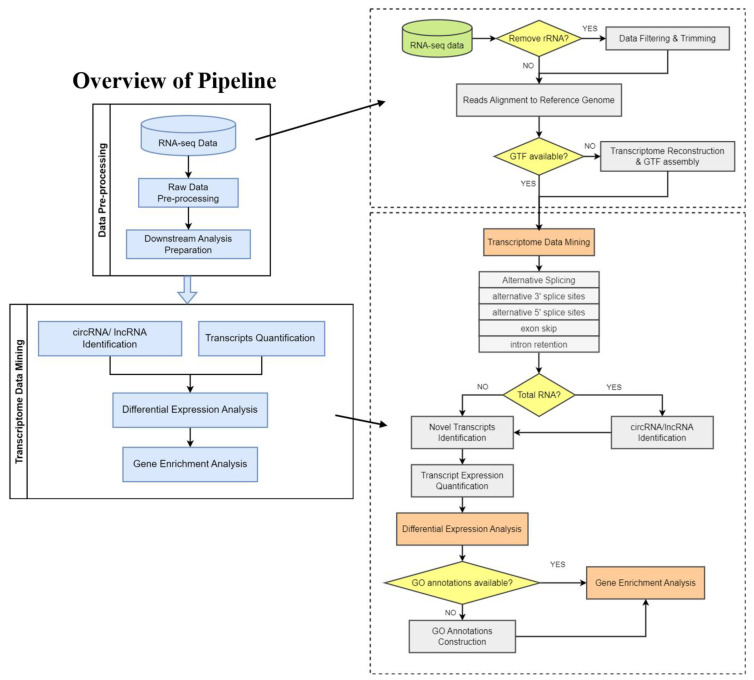
Schematic diagram of analysis workflow. PipeOne-NM was developed as a bioinformatical pipeline to perform transcriptome analyses for non-model organisms from raw RNA-seq data. PipeOne-NM integrates two modules using algorithms to assemble transcriptome, for transcriptome annotation, and for non-coding RNA detection.

**Figure 2 genes-14-00989-f002:**
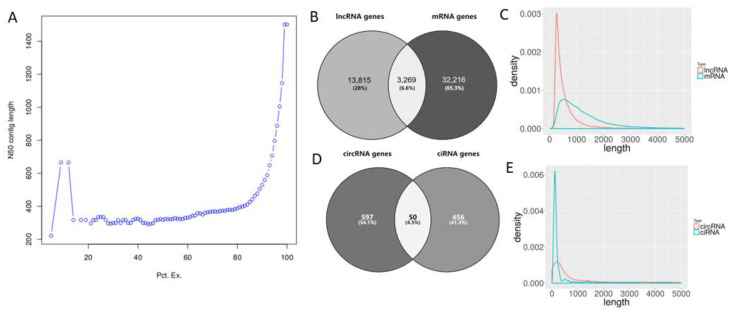
Transcriptome assembly and annotation of *Schmidtea mediterranea*. Distribution curve of transcriptome ExN50 (**A**). Venn plot showing the intersection of lncRNA and mRNA genes (**B**). Length distribution of lncRNA and mRNA (**C**). Venn plot showing the intersection of circRNA and ciRNA genes (**D**). Length distribution of circRNA and ciRNA (**E**).

**Figure 3 genes-14-00989-f003:**
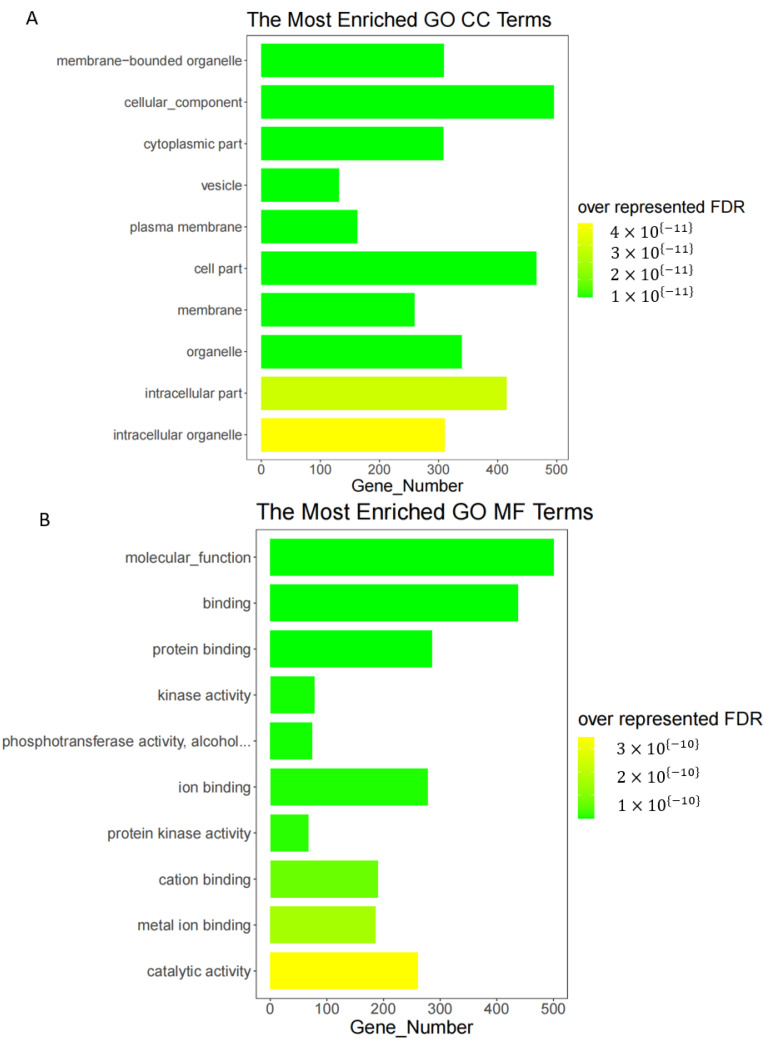
GO analysis results of ecircRNA host genes. Cellular component part of GO enrichment results of ecircRNA host genes (**A**). Molecular function part of GO enrichment results of ecircRNA host genes (**B**).

**Figure 4 genes-14-00989-f004:**
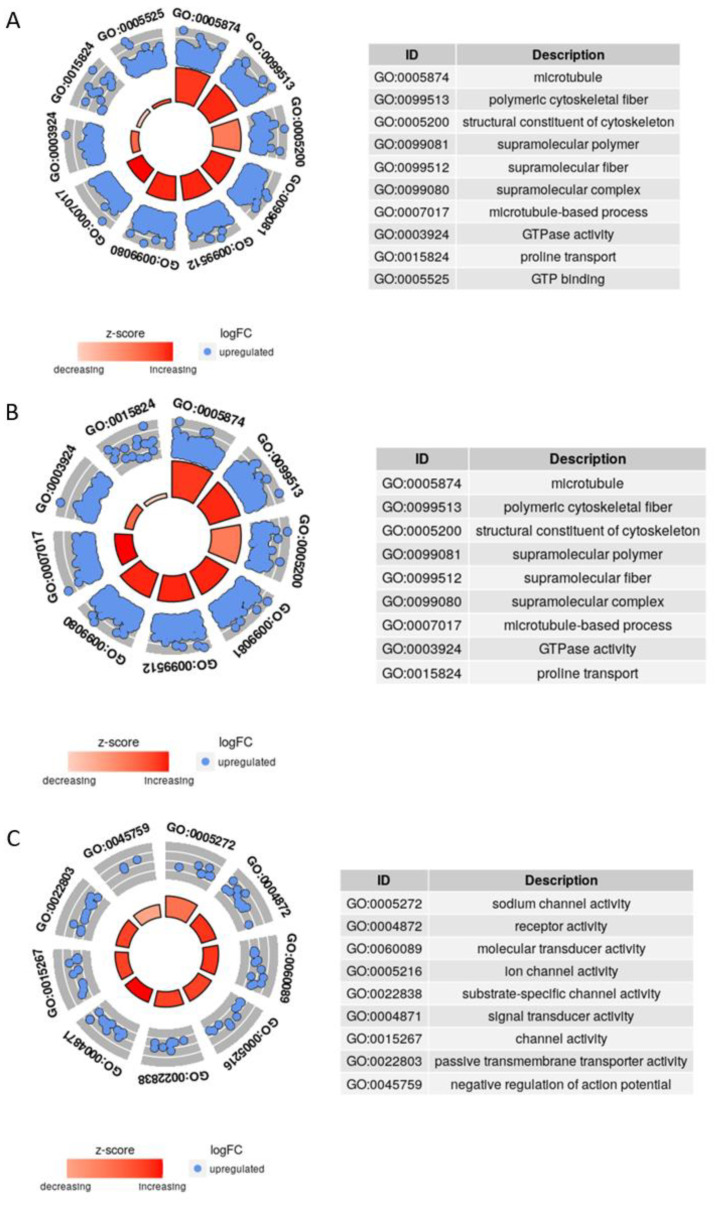
GO analysis results. GO enrichment results of up-regulated genes in mature worm of the sexual strain compared with mature worm of the asexual strain (**A**). GO enrichment results of up-regulated genes in mature sexual strain compared with juvenile of sexual strain (**B**). GO enrichment results of up-regulated genes in eye compared with whole body (**C**).

**Table 1 genes-14-00989-t001:** Required tools, version, and citations.

Tool and Resources	Versions	Citation/Download URL	Use Description
Data pre-processing:			
fasterq-dump	2.11.2	http://ncbi.github.io/sra-tools, accessed on 28 May 2022	Data format conversion.
RiboDetector	0.2.7	[37]	Remove rRNA in raw data.
fastp	0.23.0	[7]	filtering and quality controlling of reads.
HISAT2	2.2.1	[38]	Aligning sequencing reads to reference genome
samtools	1.13	[14]	Sorting reads alignment results
Stringtie	2.1.6	[15]	Reconstructing a transcriptome.
TACO	0.7.3	[16]	Merging multiple transcriptomes.
gffread	0.12.7	[39]	Extracting transcript sequences
Transcriptome data mining:			
Salmon	1.5.2	[17]	Transcript quantification.
TransDecoder	5.5.0	https://github.com/TransDecoder/TransDecoder, accessed on 28 July 2022	Prediction of open reading frame.
BLAST	2.12.0	[40]	Searching for homologous sequence
hmmscan	3.3	[22]	Searching for homologous sequence
SignalP	1.12	[23]	Prediction of signal peptide.
RNAMMER	1.2	[25]	Prediction of rRNA transcripts.
tmhmm	2.0	[24]	Prediction of transmembrane domain.
Trinnotate	3.2.2	[41]	Transcripts annotation.
CIRI2	2.0.6	[27,28]	CircRNA prediction.
BWA	0.7.17	[42]	Aligning sequencing reads to reference genome
CIRIquant	1.0	[29]	CircRNA transcripts quantification.
bedtools	2.27.1	[43]	CircRNA sequence extraction.
miRanda	3.3	[44]	CircRNA-miRNA interaction prediction.
rMATS	4.0.2	[35]	Alternative splicing analysis.
Rstudio	4.2.1	[45]	Downstream analysis platform.
goseq	1.30.0	http://bioconductor.org/packages/goseq/, accessed on 28 May 2022	Gene ontology analysis.
DESeq2	1.36.0	[30]	Gene differential expression analysis
Goplot	1.0.2	https://wencke.github.io/, accessed on 28 May 2022	Plots for gene ontology analysis.
asgal	1.1.7	[36]	Alternative splicing analysis

**Table 2 genes-14-00989-t002:** Feature comparison between PipeOne-NM and FA-nf.

	PipeOne-NM	FA-nf
Raw data processing	√	√
Quality control	√	√
Alignment	√	√
Transcriptome reconstruction	√	√
De novo transcriptome assembly	√	√
Novel lncRNA prediction	√	√
CircRNA prediction	√	×
Gene quantification	√	×
Alternative splicing	√	×
Management systems	Nextflow	-
Resume	√	×
Parallel	√	×
Docker	√	×
Conda	√	√
Singularity	√	×

**Table 3 genes-14-00989-t003:** Reference transcriptome assembly.

	Seqs	Min	Max	Average	Median	N50	GC%
Nucleotide	31,085	201	22,206	1362.98	990	1886	33.53
LS454	38,780	201	7586	746.66	591	953	34.18
CD-HIT-EST	62,562	201	22,206	1051.76	746	1433	33.73

**Table 4 genes-14-00989-t004:** Assembly of the transcriptome for annotation.

	Seqs	Min	Max	Average	Median	N50	GC%
S2	93,786	33	28,038	1308.55	1001	1703	34.21
CIW4	36,528	24	7352	555.66	424	664	33.99
Custom	4072	11	7184	662.40	486	489	33.89
CD-HIT-EST	107,317	201	28,038	1156.40	836	1593	34.22
Filtered	84,827	201	28,038	1096.67	799	1501	34.37

**Table 5 genes-14-00989-t005:** Statistics of transcriptome annotation.

	Seqs	Genes	Min	Max	Average	Median	N50	GC%
lncRNA	20,217	17,084	201	11,512	545.46	394	652	32.27
mRNA	64,582	35,485	201	28,038	1269.10	976	1623	34.65

**Table 6 genes-14-00989-t006:** Validation of identified mRNA.

Query	Number	Tool	Database	Mapped Number	Mapped Rate
Unigene_NT	32,615	tBLASTX	Trinotate_mRNA	27,789	85.20
Trinotate_mRNA	64,582	tBLASTX	Unigene_NT	64,024	99.14
Unigene_AA	32,615	tBLASTN	Trinotate_mRNA	27,420	84.07
Trinotate_mRNA	64,582	BLASTX	Unigene_AA	64,150	99.33

**Table 7 genes-14-00989-t007:** Statistics of circRNA detection.

	Seqs	Genes	Min	Max	Average	Median	N50	GC%
ecircRNA	1308	647	48	99,345	5970.84	704	24,968	26.82
ciRNA	2173	506	43	204,158	12,510.95	183	61,846	29.00
Total	3481	1103	43	204,158	10,053.48	287	61,580	28.51

## Data Availability

Not applicable.

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
