# Peer review of "Comprehensive RNA-Seq Analysis Pipeline for Non-Model Organisms and Its Application in Schmidtea mediterranea"

_genes, 2023, doi:10.3390/genes14050989_

Round 1
Reviewer 1 Report
The manuscript has two parts
Description of a bioinformatics application designed to facilitate the processing and analysis of RNA-seq data, with special emphasis on its applicability to non-model organisms.
Use of the described application for a series of transcriptomic data analysis of Schmidtea mediterranea.
Positive aspects of the PipeOne-NM system description are:
It is relatively easy to install, using standard containers, without the need for special server configuration.
It covers all steps, from the processing of raw sequences to the annotation of non-model organisms, using methods well accepted by the scientific community. It even allows the analysis of circular RNAs (circRNAs).
The result files are saved in a well-organized way for each processing/annotation step.
The negative aspects of the PipeOne-NM system description are:
In a sense it is a black box whose only configuration options serve to indicate the path of files and databases to be used. This limits its use in complex cases or with designs different from the one used to illustrate the use of the system.
Although presented as a one-step RNA-seq analysis pipeline, the final part of the process (differential expression analysis) is presented as a separate script to be handled separately.
Neither the manuscript nor the tool's web page mentions the RAM, CPU and hard disk requirements needed to install and run the application depending on the complexity of the organism to be analyzed.
The proposed metric for assessing the quality of transcriptome assembly (N50) is not very adequate (https://bmcbioinformatics.biomedcentral.com/articles/10.1186/s12859-020-3382-4).
Although the tool is useful for unannotated organisms, the genome sequence is required, which limits its applicability to many species. In practice, once an organism has been sequenced, its (at least preliminary) annotation usually accompanies the sequence in reference databases.
The positive aspects of the analysis of Schmidtea mediterranea are:
They have collected a large amount of raw RNA-seq data for that organism (237 samples) under well-defined physiological conditions (sexual strain/asexual strain mature/young, different body parts) that allow comparative analyses of many types.
The negative aspects of the analysis of Schmidtea mediterranea are:
The exposition of the results obtained is unclear: FDR values based on differential gene expression are mixed with FDR values relative to GO terms. In fact, in many cases, the FDRs of the differentially expressed genes are very poor and the subsequent functional analysis based on these listings is neither conclusive nor provides any novelty (e.g., in point 3.4 relevant GO terms are listed for circRNAs that, despite being very generic, the authors claim to be consistent with function as miRNA sponges).
Moreover, this part of the results is presented with hardly any figures (MA plots or similar) to help assess the quality of the filtered gene listings, on which the conclusions based on their functional analysis depend.
Author Response
Point 1: In a sense it is a black box whose only configuration options serve to indicate the path of files and databases to be used. This limits its use in complex cases or with designs different from the one used to illustrate the use of the system.
Although presented as a one-step RNA-seq analysis pipeline, the final part of the process (differential expression analysis) is presented as a separate script to be handled separately.
Response 1: Thank you for pointing out this problem. PipeOne-NM was originally designed to provide a one-stop tool for RNA-seq analysis of non-model organisms, reducing the code knowledge required of the user, so it will be somewhat of a black box. For researchers who have knowledge of code and wish to apply PipeOne-NM in complex studies, they can personalize the output files of each step of PipeOne-NM, allowing the analysis tool to be applied to different complex situations. Again, we will be adding more features to PipeOne-NM as needed to make it more widely applicable in the future.
As for the problem that the script for differential analysis is independent of PipeOne-NM and requires additional manipulation, we believe that each step of the differential analysis and the corresponding parameters need to be considered and evaluated by the operator/user, like considering about the dispersion parameter of the sequencing data, which two treatment conditions need to be compared for differential gene expression, and what is the cutoff of log fold change and p-value. These analyses need to be tailored for each study and are difficult to achieve through a standardized process.
Point 2: Neither the manuscript nor the tool's web page mentions the RAM, CPU and hard disk requirements needed to install and run the application depending on the complexity of the organism to be analyzed.
The proposed metric for assessing the quality of transcriptome assembly (N50) is not very adequate (https://bmcbioinformatics.biomedcentral.com/articles/10.1186/s12859-020-3382-4).
Although the tool is useful for unannotated organisms, the genome sequence is required, which limits its applicability to many species. In practice, once an organism has been sequenced, its (at least preliminary) annotation usually accompanies the sequence in reference databases.
Response 2: We thank this reviewer for pointing out these problems in the manuscript. As for the RAM and CPU requirements, we add the description on the tool’s web page: CPU 33 and RAM 55G.
In transcriptome assembly, the N50 value is used to assess the contiguity and completeness of the assembly. A higher N50 value indicates that the assembly is more contiguous, with longer contigs or transcripts, and is likely to represent more complete and accurate transcripts. Although the N50 value alone is not sufficient to evaluate the quality of a transcriptome assembly, the Unigene alignment result in the manuscript can also indicate the quality of transcriptome assembly.
Admittedly, one of the current limitations of PipeOne-NM is that it requires the input of reference genome of a non-model organism, which hampers the application of PipeOne-NM in non-model species without available reference genome. In the future maintenance and refinement of PipeOne-NM, we will remove the mandatory requirement for species reference genomes and include a module to build reference genomes from scratch, expanding the species application of PipeOne-NM.
Point 3: The exposition of the results obtained is unclear: FDR values based on differential gene expression are mixed with FDR values relative to GO terms. In fact, in many cases, the FDRs of the differentially expressed genes are very poor and the subsequent functional analysis based on these listings is neither conclusive nor provides any novelty (e.g., in point 3.4 relevant GO terms are listed for circRNAs that, despite being very generic, the authors claim to be consistent with function as miRNA sponges).
Moreover, this part of the results is presented with hardly any figures (MA plots or similar) to help assess the quality of the filtered gene listings, on which the conclusions based on their functional analysis depend.
Response 3: Thank you for pointing out these problems in the manuscript. It is important to acknowledge that the downstream differential analysis in this manuscript does not delve into the biological significance of the transcriptome information in Schimidtea mediterranea, and we believe that the purpose of this manuscript is to introduce PipeOne-NM, a tool for non-model organism transcriptome analysis, and therefore we believe that in future maintenance and refinement of PipeOne-NM, appropriate modules will be added to make the pipeline more comprehensive.
Reviewer 2 Report
I commend the authors for improving the manuscript. I think the article reads well now and it could prove a valuable resource for the community studying non-model organisms. I have no queries from my end.
Author Response
We thank the reviewer for these comments.
Reviewer 3 Report
This manuscript describes a thorough transcriptome assembly and analysis pipeline and it's application to Schmidtea mediterranea. The pipeline software is freely available and incorporates widely adopted open source analysis tools for specific steps in the pipeline. One aspect that would strengthen the paper is to include a comparison of the PipeOne-NM pipeline applied to a well studied genome.
I recommend applying the pipeline to a well-studied non-vertebrate genome such as C. elegans. It would be interesting to compare the performance of the pipeline on various types of non-coding RNA annotations. How many of each class of RNA are captured by the pipeline?
Author Response
Point 1: One aspect that would strengthen the paper is to include a comparison of the PipeOne-NM pipeline applied to a well-studied genome.
Response 1:
This is a very reasonable suggestion. Since PipeOne-NM is built for non-model organisms, it is mainly used for annotation construction and relevent analyses of non-model organisms. And we recommend using PipeOne (DOI: 10.3390/genes12121865) for analysis of fully annotated organisms with a well-constructed reference genome, so we have not included PipeOne-NM for fully annotated species in this manuscript. And the corresponding descriptions have been added to the manuscript.
Point 2: I recommend applying the pipeline to a well-studied non-vertebrate genome such as C. elegans. It would be interesting to compare the performance of the pipeline on various types of non-coding RNA annotations. How many of each class of RNA are captured by the pipeline?
Response 2:
We thank the reviewer for this suggestion. PipeOne-NM can identify mRNAs, circRNAs and lncRNAs in the RNA-seq data of the non-model organisms. while it is interesting to apply PipeOne-NM to well-studied species (e.g. C. elegans) to compare the annotation of different types of non-coding RNAs, doing so does not seem to guarantee the same conclusions in other non-model species. Therefore, we did not cover similar content in our manuscript. However, in the future maintenance and refinement of PipeOne-NM, we believe it is important to increase the annotation efficiency and accuracy of non-coding RNAs. Then we can compare the annotation efficiency of different types of non-coding RNAs using improved and pre-improved PipeOne-NM analysis on the same set of data, so that the conclusions drawn would be more broadly applicable.
Reviewer 4 Report
In this work, Wanh et al. proposed an integrated RNA-seq analysis pipeline for transcriptome functional annotation, non-coding RNA identification, and transcripts alternative splicing analysis of non-model organisms. The proposed pipeline is demonstrated on Schmidtea mediterranea and shows promising results for transcriptome assembly and identifications in mRNA, lncRNA, and in circRNAs. However, there are a few issues/open questions in the manuscript that I believe should be addressed.
1. For tools selection, it is a little surprising to see the authors choose BWA for fastq alignment. BWA has not been updated in a long time and newer aligners such as STAR(PMID: 23104886) provide much faster speed with comparable(if not better) performance. Similar to transcriptome assembly. I believe this selection may become an obstacle for others considering the use of the proposed tool. Can the author elaborate on the choice of tool for the pipeline? Can they be substituted by more recent approaches?
2. Another restriction for others (like me) to consider using the pipeline is that it seems the results are only tested on a limited dataset of non-model organisms. Did the author consider using other organisms for similar experiments described for Schmidtea mediterranea? How would the pipeline perform when generalized to a broad range of non-model organisms?
3. It is nice to see co-expression analysis is also included in the pipeline. However, we know that such analysis might be susceptible to false positives from indirect links of genes. I believe that it would be beneficial to incorporate gene/pathway network modeling approaches as well. Example of such approaches includes ARACNe(PMID: 17406294)/ GENIE3 (PMID: 20927193), or the authors can compare the performance of the current approach with those tools for a quick benchmark.
Author Response
Point 1: For tools selection, it is a little surprising to see the authors choose BWA for fastq alignment. BWA has not been updated in a long time and newer aligners such as STAR(PMID: 23104886) provide much faster speed with comparable(if not better) performance. Similar to transcriptome assembly. I believe this selection may become an obstacle for others considering the use of the proposed tool. Can the author elaborate on the choice of tool for the pipeline? Can they be substituted by more recent approaches?
Response 1: Thank you for pointing out this problem. In fact, BWA is selected based on the input file needs of the circRNA detection software CIRI2 in PipeOne-NM. In addition to the circRNA detection module, HISAT2 was selected in PipeOne-NM for reference genome alignment of sequencing data. In the future maintenance and refinement of PipeOne-NM, we will include more options for software at each step, in the hope of providing a more tailored solution for future studies of non-model species.
Point 2: Another restriction for others (like me) to consider using the pipeline is that it seems the results are only tested on a limited dataset of non-model organisms. Did the author consider using other organisms for similar experiments described for Schmidtea mediterranea? How would the pipeline perform when generalized to a broad range of non-model organisms?
Response 2: We thank the reviewer for these suggestions. In this manuscript, we applied PipeOne-NM to the analysis of 237 Illumina platform-based RNA seq runs of the Schimidtea mediterranea. And we hold the view that the application in a dataset of such size can reflect the performance of PipeOne-NM. In addition, the software chosen for each step of the PipeOne-NM is one that has been tested over a long period of time to provide reliable results, and therefore we are confident that PipeOne-NM will work as well in the analysis of data from other non-model species as it does in the Schimidtea mediterranea.
Point 3: It is nice to see co-expression analysis is also included in the pipeline. However, we know that such analysis might be susceptible to false positives from indirect links of genes. I believe that it would be beneficial to incorporate gene/pathway network modeling approaches as well. Example of such approaches includes ARACNe (PMID: 17406294)/ GENIE3 (PMID: 20927193), or the authors can compare the performance of the current approach with those tools for a quick benchmark.
Response 3: Thank you for this reasonable suggestion. In future maintenance and refinement of PipeOne-NM, we will incorporate relevant gene/pathway network approaches to the lncRNA-mRNA co-expression analysis section of pipeline for the reduction of false positive.
Reviewer 5 Report
Wang et al. have developed a comprehensive, integrated pipeline for the analysis of Illumina cDNA reads from non-model organisms. They build on existing workflows that have been developed by other researchers (e.g. Trinity, DESeq2, EdgeR), bringing them all together into a single docker container (or conda environment, depending on user preference), substantially reducing the researcher effort for data processing. To demonstrate how their workflow works, they have fed a corpus of Schmidtea mediterranea datasets into their pipeline.
My biggest concern with the work demonstrated in this manuscript is that the outputs of the research from the example datasets are not being made available. I would be much more comfortable with recommending publication if these results were published alongside the paper, but acknowledge that the example datasets are used mostly to demonstrate that the pipeline works, rather than for their intrinsic value.
Regardless, I think that the presentation of this pipeline in the manuscript is excellent. I have a few suggestions for improvement, mentioned below:
Lines 109-112: "6 LS454 platform-based single-end RNA-seq runs... were used to de novo construct reference transcriptome via Trinity (version 2.6.5)"
- This is not the most recent Trinity paper; a better reference (especially because L170-171 indicates downstream analyses are being run) would be Haas et al., 2013
- Given that there is an existing genome for Smed (i.e. schMEDS2-may2021), and reference genomes are a required component of your pipeline, why was Trinity's genome-guided
transcriptome assembly mode not used?
Lines 112-114: "After two sets of transcriptomes were integrated,... were used to assemble reference transcriptome"
- This seems like it would be an incredibly useful resource for Smed researchers; where can this merged transcriptome be found? Has it been added to SmedGD or PlanMine?
Line 148-150: "we downloaded the Unigene sequences of Schmidtea mediterranea from the SmedGD"
- Why is SmedGD being used here, and not PlanMine [https://planmine.mpibpc.mpg.de/planmine/report.do?id=2000001#ad-image-0]?
Lines 173-174: "Genes with a threshold of more than 4-fold change in expression levels between the two groups of samples and P < 0.001"
- is this adjusted or unadjusted P values? Was any fold change shrinkage carried out?
Lines 260-261: "The change of length of N50 in Schmidtea mediterranea indicate that longer transcripts were tended to expressed at a lower level while shorter transcripts were tended to express at a higher level"
- Are these numbers from before or after length-based shot-noise correction (e.g. variance-stabilised transformation using DESeq2, or RPKM if VST cannot be carried out)
Lines 278-279: "The maximum length of lncRNA sequence length was less than half that of mRNA, indicating that the length of mRNA is longer in Schmidtea mediterranea"
- I suspect something else is going on here. If (for example) most of the sequencing was polyA selection, then there would be a bias towards mRNA in the detected transcripts
There are numerous minor English readability issues scattered throughout the manuscript. It would be useful to have someone scan the paper for readability and understandability. Here are a few examples (not an exhaustive list):
- L28 "profiles. And samples from different" -> "profiles. Samples from different"
- L59 "Some packages have been developed to realize aforementioned functions already" -> "Some packages have been developed that carry out these functions already"
- L354 "Because Schmidtea mediterranea do not have the organ liver" -> "Because Schmidtea mediterranea do not have a liver"
- L360-361 "comprehensive transcriptome analysis pipeline non-model organisms" -> "comprehensive transcriptome analysis pipeline for non-model organisms"
Additional comments:
Given that there are only 17 bioprojects in the downloaded data, I think it would be a good idea to include references to papers associated with these projects, where available. Citing their research helps to demonstrate the value of the datasets (which have clearly been useful for you) to future researchers and funders. This would mirror the good practice you have demonstrated by citing papers associated with the programs used in the pipeline.
It seems like you have done your own independent development of this pipeline, and have only used the Schmidtea mediterranea genome as an exemplar to demonstrate the use of your pipeline. However, it would be a bit of a waste to leave this resource hidden away. I recommend that you consider contacting the curators of the planarian database to see if any of the valuable resources you have created can be added to their database. In the very least, a citation to PlanMine would be helpful [https://dx.doi.org/10.1093/nar/gky1070]
The manuscript states "The datasets supporting the conclusions of this article are included within the article and its additional files." However, no datasets are provided, apart from a non-published table of datasets. Given that the authors have clearly produced a number of interesting resources (e.g. annotated / merged transcriptome, differential expression lists), it seems a waste to not publish these alongside the paper (or in an associated Zenodo archive).
Additional non-manuscript comments:
* The docker image available at sijunli/pipeone_nm:latest does not have any Dockerfile available on dockerhub, and the image layer text is not very helpful. Please include a Dockerfile in the github repository, or on dockerhub, so that it is a little bit easier for people to help develop this work.
* The github repository has no version tags, yet the paper states version 0.0.1
* It would be appreciated if an eternal citation/DOI could be created for the code linked to the publication (e.g. by using Github's links to Zenodo)
* Due to time constraints for MDPI reviews, I have not run the pipeline myself to evaluate its effectiveness for my own Schmidtea mediterranea data
Author Response
Point 1: Lines 109-112: "6 LS454 platform-based single-end RNA-seq runs... were used to de novo construct reference transcriptome via Trinity (version 2.6.5)"
- This is not the most recent Trinity paper; a better reference (especially because L170-171 indicates downstream analyses are being run) would be Haas et al., 2013
- Given that there is an existing genome for Smed (i.e. schMEDS2-may2021), and reference genomes are a required component of your pipeline, why was Trinity's genome-guided transcriptome assembly mode not used?
Response 1: Thank you for pointing out this problem. We have changed the reference in the manuscript accordingly. As for the problem of not using Trinity’s genome-guided transcriptome assembly mode, that’s because in Trinity’s genome-guided assembly, the genome is used to partition the reads according to locus. However the reference genome with chromosome-division was not yet available before we finished the upstream works. Since that, the genome-guided mode of Trinity was not used.
Point 2: Lines 112-114: "After two sets of transcriptomes were integrated,... were used to assemble reference transcriptome"
- This seems like it would be an incredibly useful resource for Smed researchers; where can this merged transcriptome be found? Has it been added to SmedGD or PlanMine?
Response 2: It’s a reasonable suggestion. But as the focus of this manuscript is on our development of PipeOne-NM, a tool for the transcriptome analysis of non-model organisms, we would prefer readers to focus on the tool itself, and the related resources of Schmidtea mediterranea will be uploaded to PlanMine or SmedGD after further refinement.
Point 3: Line 148-150: "we downloaded the Unigene sequences of Schmidtea mediterranea from the SmedGD"
- Why is SmedGD being used here, and not PlanMine [https://planmine.mpibpc.mpg.de/planmine/report.do?id=2000001#ad-image-0]?
Lines 173-174: "Genes with a threshold of more than 4-fold change in expression levels between the two groups of samples and P < 0.001"
- is this adjusted or unadjusted P values? Was any fold change shrinkage carried out?
Lines 260-261: "The change of length of N50 in Schmidtea mediterranea indicate that longer transcripts were tended to expressed at a lower level while shorter transcripts were tended to express at a higher level"
- Are these numbers from before or after length-based shot-noise correction (e.g. variance-stabilised transformation using DESeq2, or RPKM if VST cannot be carried out)
Lines 278-279: "The maximum length of lncRNA sequence length was less than half that of mRNA, indicating that the length of mRNA is longer in Schmidtea mediterranea"
- I suspect something else is going on here. If (for example) most of the sequencing was polyA selection, then there would be a bias towards mRNA in the detected transcripts
Response 3: We thank the reviewer for pointing out these problems. It is important to acknowledge that we did not use sequences from the PlanMine database when collecting the data, but only downloaded sequences from the Smed database, which we will add information from PlanMine in future studies.
The P-values are adjusted P-values. And the FC shrinkage has not been done according to the Trinity downstream analysis pipeline.
Lines 260-261: "The change of length of N50 in Schmidtea mediterranea indicate that longer transcripts were tended to expressed at a lower level while shorter transcripts were tended to express at a higher level" The conclusion are drawn based on the TMM value of transcripts, not from before or after length-based noise correction.
Lines 278-279: "The maximum length of lncRNA sequence length was less than half that of mRNA, indicating that the length of mRNA is longer in Schmidtea mediterranea". We found that this sentence was not phrased properly and we have corrected the sentence as " The maximum length of mRNA is greater than lncRNA "
Point 4:
There are numerous minor English readability issues scattered throughout the manuscript. It would be useful to have someone scan the paper for readability and understandability. Here are a few examples (not an exhaustive list):
- L28 "profiles. And samples from different" -> "profiles. Samples from different"
- L59 "Some packages have been developed to realize aforementioned functions already" -> "Some packages have been developed that carry out these functions already"
- L354 "Because Schmidtea mediterranea do not have the organ liver" -> "Because Schmidtea mediterranea do not have a liver"
- L360-361 "comprehensive transcriptome analysis pipeline non-model organisms" -> "comprehensive transcriptome analysis pipeline for non-model organisms"
Response 4: We thank this reviewer for pointing out the misleading sentences and typos in this manuscript. According to your suggestion, we have rechecked the relevant wording and sentence in the manuscript and corrected the incorrect expressions.
Point 5: * The docker image available at sijunli/pipeone_nm:latest does not have any Dockerfile available on dockerhub, and the image layer text is not very helpful. Please include a Dockerfile in the github repository, or on dockerhub, so that it is a little bit easier for people to help develop this work.
* The github repository has no version tags, yet the paper states version 0.0.1
Response 5: We thank this reviewer for the suggestion. Actually, constructing docker images with dockerfile often fails due to network speed. so when constructing the docker image in sijunli/pipeone_nm:latest, the interaction mode is utilized and the dockerfile will not be attached.
We correct the misleading version tag mentioned in the manuscript, in consistent with the web page of PipeOne-NM.
Round 2
Reviewer 1 Report
Thank you for answering my questions.
Unfortunately, I think the manuscript has not been improved enough. My issues had more to do with the type of development that, in my opinion, does not solve the task of carrying out an RNA-Seq protocol in any non-model organism. It has little applicability to genomes other than the one used in the manuscript and part of the steps needed to complete the described analysis are done with additional tools not included in the pipeline. The prerequisites and the complexity of adjusting the parameters to other data sets defeats the purpose of presenting a pipeline that facilitates the work of analyzing the differential expression of other non-model organisms.
Regarding the need to detail the computational requirements, I do not understand what "CPU: 33" means.
Regarding the problem of differential expression results, it has not been adequately answered. These results occupy an important part of the manuscript and are intended to serve to demonstrate the usefulness of the developed pipeline, but they are still presented in a confusing way.
I believe that the way to improve the manuscript is precisely by improving the presentation of the biological results obtained with Schmidtea, putting the focus on them, mentioning the computational tools used to obtain them in the Materials and Methods section.
Author Response
Point 1: My issues had more to do with the type of development that, in my opinion, does not solve the task of carrying out an RNA-Seq protocol in any non-model organism. It has little applicability to genomes other than the one used in the manuscript and part of the steps needed to complete the described analysis are done with additional tools not included in the pipeline. The prerequisites and the complexity of adjusting the parameters to other data sets defeats the purpose of presenting a pipeline that facilitates the work of analyzing the differential expression of other non-model organisms.
Response 1: We thank this reviewer for these comments, which focuses on the part of differential expression analysis for non-model organisms that is not integrated into PipeOne-NM, and the subsequent cumbersome parameter adjustment somehow defeats the purpose of a one-stop analysis. However we believe that without adjusting parameters on a species-specific basis, the differentially expressed genes identified from such analyses may have no real biological significance and may not truly guide research in non-model organisms. Because there are so many factors and indicators to consider in differential gene expression analysis, such as the dispersion and the batch effect of the data, these parameters need to be specifically considered in downstream analyses before meaningful biological conclusions can be drawn. PipeOne-NM can give researchers greater analytical freedom in the downstream analysis while reducing the number of steps required to build upstream analyses.
Point 2: Regarding the need to detail the computational requirements, I do not understand what "CPU: 33" means.
Response 2: We thank this reviewer for pointing out misleading expressions in the website of PipeOne-NM. The ‘CPU: 30’ means that PipeOne-NM needs 30 CPU cores when processing the data. We have made corresponding corrections in the website.
Point 3: Regarding the problem of differential expression results, it has not been adequately answered. These results occupy an important part of the manuscript and are intended to serve to demonstrate the usefulness of the developed pipeline, but they are still presented in a confusing way.
Response 3: Thank you for pointing out these problems in the manuscript. Due to the number of biological conditions involved in the RNA-seq data collected, that it was difficult to focus on a specific biological problem of Schimidtea mediterranea, moreover, in order to fully demonstrate the effectiveness of PipeOne-NM, we present the results of the analysis of different data set, so in the differential analysis section we present the GO enrichment results for the sexual and asexual strain, juvenile and mature and different body parts of Schimidtea mediterranea.
Reviewer 3 Report
I'm recommend publication in its current form, as non-coding annotations are not my specialty. I'll be looking into the robustness of automated non-coding annotation software for projects I'm involved in and will keep your pipeline in mind.
Author Response

(The authors gave the same response as above.)

Reviewer 4 Report
I would like to express my gratitude to the authors for their substantial efforts in addressing my comments. It is evident that the manuscript has undergone significant improvement since the revision. While there are no major concerns regarding the current form of the manuscript, I do believe that additional evidence is necessary to further substantiate the proposed pipeline beyond the single species validated in the experiment (Schmidtea mediterranea). Furthermore, I would like to suggest that the authors review the manuscript thoroughly for any text writing and formatting issues, including the accidental addition of foreign characters in the header of the reference section.
Author Response
We thank the reviewer for these comment and have checked the formatting issues of the manuscript in the light of the reviewer's comments.